# Designing highly efficient interlocking interactions in anisotropic active particles

Solenn Riedel[1], Ludwig A. Hoffmann ®[2], Luca Giomi ®[2] & Daniela J. Kraft ®[1] ✉

Cluster formation of microscopic swimmers is key to the formation of biofilms and colonies, efficient motion and nutrient uptake, but, in the absence of other interactions, requires high swimmer concentrations to occur. Here we experimentally and numerically show that cluster formation can be dramatically enhanced by an anisotropic swimmer shape. We analyze a class of model microswimmers with a shape that can be continuously tuned from spherical to bent and straight rods. In all cases, clustering can be described by Michaelis-Menten kinetics governed by a single scaling parameter that depends on particle density and shape only. We rationalize these shape-dependent dynamics from the interplay between interlocking probability and cluster stability. The bent rod shape promotes assembly in an interlocking fashion even at vanishingly low particle densities and we identify the most efficient shape to be a semicircle. Our work provides key insights into how shape can be used to rationally design out-of-equilibrium self-organization, key to creating active functional materials and processes that require two-component assembly with high fidelity.

The motility inherent to synthetic[1-3] and living[4-6] microswimmers is crucial for their collective behavior. Besides pushing these systems out of equilibrium, it is at the heart of the occurrence of trains[7,8], flocks[3,9], vortices[9,10], and clusters[1,2,4-6,11]. Cluster formation occurs in a range of active agents, from self-propelled colloids forming living active crystals to rotating or moving clusters of populations of starfish embryos[6], or motile bacteria like *Thiovulum majus*[5] and *Myxococcus xanthus* mutants[4]. There, it is thought to be important for the formation of colonies and biofilms, efficient motion, and nutrient uptake[12].

Simulations of self-propelled agents have identified that motility is a minimal requirement for cluster formation, and that it occurs even in systems of repulsive agents at sufficiently high densities. This so-called *motility induced phase separation* (MIPS)[11,13-16] originates from a slowing-down of the active agents upon collision. Systems of self-propelled disks at high particle densities phase separate above a critical density of 40%[17], forming a single, globally disordered, macroscopic MIPS-aggregate that experiences diffusive motion. In experiments with synthetic microswimmers, however, cluster formation usually sets in already at several percent due to the presence of

additional attractive interactions, and multiple disordered and dynamic aggregates are found across the sample[1,2,18].

In contrast to synthetic active particles, biological microswimmers often have non-spherical shapes[19-22] and simulations have predicted that their collective behavior is strongly influenced by shape. For example, longitudinally-propelled rods self-organize into polar bands instead of clusters as a consequence of the lateral association promoted by their elongated shape[23-26]. Clusters of anisotropic active particles may not only display Brownian diffusion, but also directed and spinning motion[27,28]. In addition, the critical density at which cluster form has been proposed to depend on shape because it strongly influences inter-particle slowing down[28]. For active hexagons, for example, efficient deceleration upon collision results in nucleation of many small clusters at far lower particle densities than would be expected for spheres[28].

Despite these predictions, there is little experimental work due to the limited availability of active particles with an anisotropic shape. Experiments with transversely-propelled rods revealed that stable doublets already form at particle densities of ≈1%[29] and clustering is

[1]Soft Matter Physics, Huygens-Kamerlingh Onnes Laboratory, Leiden University, PO Box 9504, 2300 RA Leiden, The Netherlands. [2]Instituut-Lorentz, Leiden University, P.O. Box 9506, 2300 RA Leiden, The Netherlands. ✉e-mail: kraft@physics.leidenuniv.nl

favored with increasing aspect ratio of the rods[29,30]. Tori, both horizontally and vertically oriented, were observed to form dynamic unstable clusters[31] and standing disks were found to cluster at about 10%[32]. While more complex shapes have been prepared[33,34] it is often challenging to selectively generate sufficient quantities to test their phase behavior and typically not possible to gradually tune their shape to find optimal clustering conditions.

Here, we use 3D micro-fabrication to create active particles with a shape that is continuously tunable between a sphere and a rod, i.e. bent rods, and study their collective behavior at low particle densities. We find that clustering occurs at extremely low surface area fractions (below 0.022%) due to a highly efficient nucleation process that relies on interlocking between the particles. Complementing our experiments with simulations and an analytical model, we find that the self-organization at low particle densities can be captured by a Michaelis-Menten kinetics and thus can be characterized by a single scaling parameter that depends on the particle density and shape only. We demonstrate that the efficiency of the self-organization process is strongly influenced by the concave shape, which affects the interplay between interlocking and stability, and show that the combination of directed motion and steric interactions leads to motility induced clustering. Our insights provide a generic understanding of how phase separation occurs in a whole class of anisotropic particles from spheres to rods as well as a strategy to precisely control the stability of active particles through the shape of their interaction site, thus enabling the rational design of their assembly pathways into functional active materials.

## Results

We exploit 3D microprinting based on two-photon polymerization to create active particles with a tunable anisotropic shape, see the Methods section for more details on particle fabrication. We print bent rods with varying opening angle $\alpha$ and constant cross-section $L = 10\,\mu m$ as this shape interpolates smoothly between a sphere and a rod, see Fig. 1a, b[35]. We render the particles active by sputter coating them on their convex side with a 5 nm thick Pt/Pd (80/20) layer and dispersing them in aqueous hydrogen peroxide ($H_2O_2$) solution. Their self-propulsion is driven by solute gradients generated through a catalytic decomposition of $H_2O_2$ at the Pt/Pd cap[36]. Depending on the fuel concentration, the bent rods either swim with their concave (1% $H_2O_2$) or convex side (5% $H_2O_2$) leading, see Supplementary Videos 1–3. Due to their size, the motion of these active crescents shows long persistence lengths, see Fig. 1c.

### Collective behavior of active bent rods

We start by examining the collective behavior of active bent rods that swim with their concave-side leading, have an opening angle of $\alpha = 180°$ and an average velocity of $\langle v \rangle_{180} = 0.78 \pm 0.08\,\mu m\,s^{-1}$. Small rotating clusters and pairs of particles quickly form after sedimentation, making our system effectively 2D, see Fig. 2a, b and Supplementary Videos 4-6. Simulations using a minimal model that only contains two ingredients, i.e., self-propelling bent rods and steric interactions between particles[37], also show the quick formation of stable pairs and rotating clusters (Fig. 2c and Supporting Video 13). Clearly, the concave shape of the active particles is crucial as it promotes interlocking and stability against breakup.

Remarkably, stable interlocking occurs already for only two particles upon a head-on collision. The translational motion of the individual particles is then transformed into a stationary rotation of the pair, see Fig. 2d. The direction of rotation is determined by the torque created by the initial off-center alignment of the particles and remains stable over the duration of our experiment, with a typical angular velocity of $\approx 0.13 \pm 0.01\,rad\,s^{-1}$. This angular velocity is in line with the active forces at play, which are in the range of $F_{active} = 0.04–0.1$ pN (see SI).

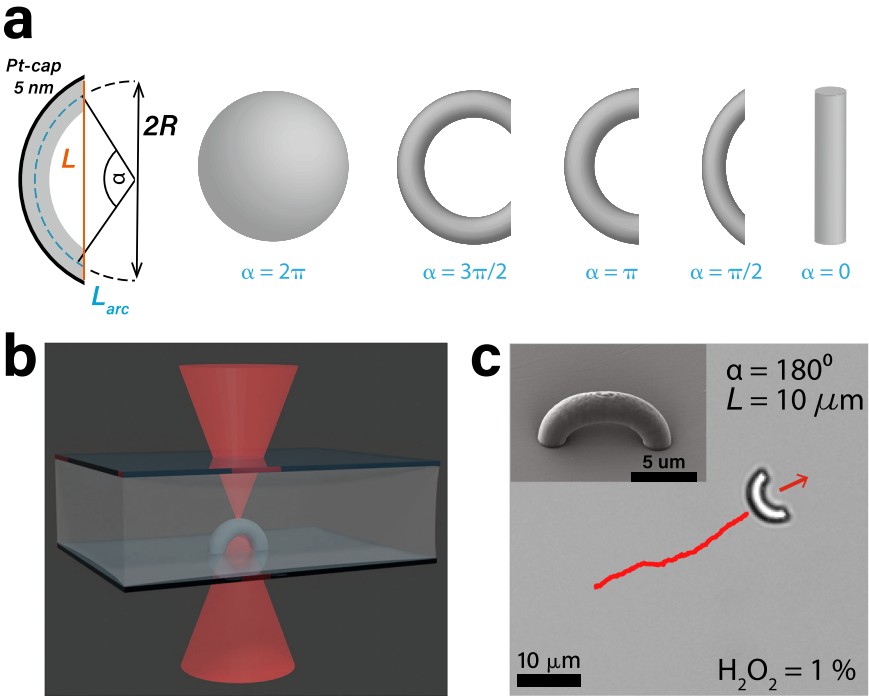

**Fig. 1 | Experimental design of self-propelled bent rods. a** By changing the design parameters, the shape of the anisotropic microswimmers can be continuously tuned from a sphere to a bent or straight rod. Cross-sectional length $L$, arc-length $L_{arc}$, opening angle $\alpha$, and radius of curvature $R$. **b** Schematic illustration of 3D printing of a bent rod by two-photon polymerization. **c** Scanning electron microscopy image (top left inset) and bright-field image of a 3D printed active crescent with $\alpha = 180°$, see the Methods section for details on particle imaging. The particle moves, concave-side leading, in the direction of the red arrow. Red line is the trajectory of past 30 s.

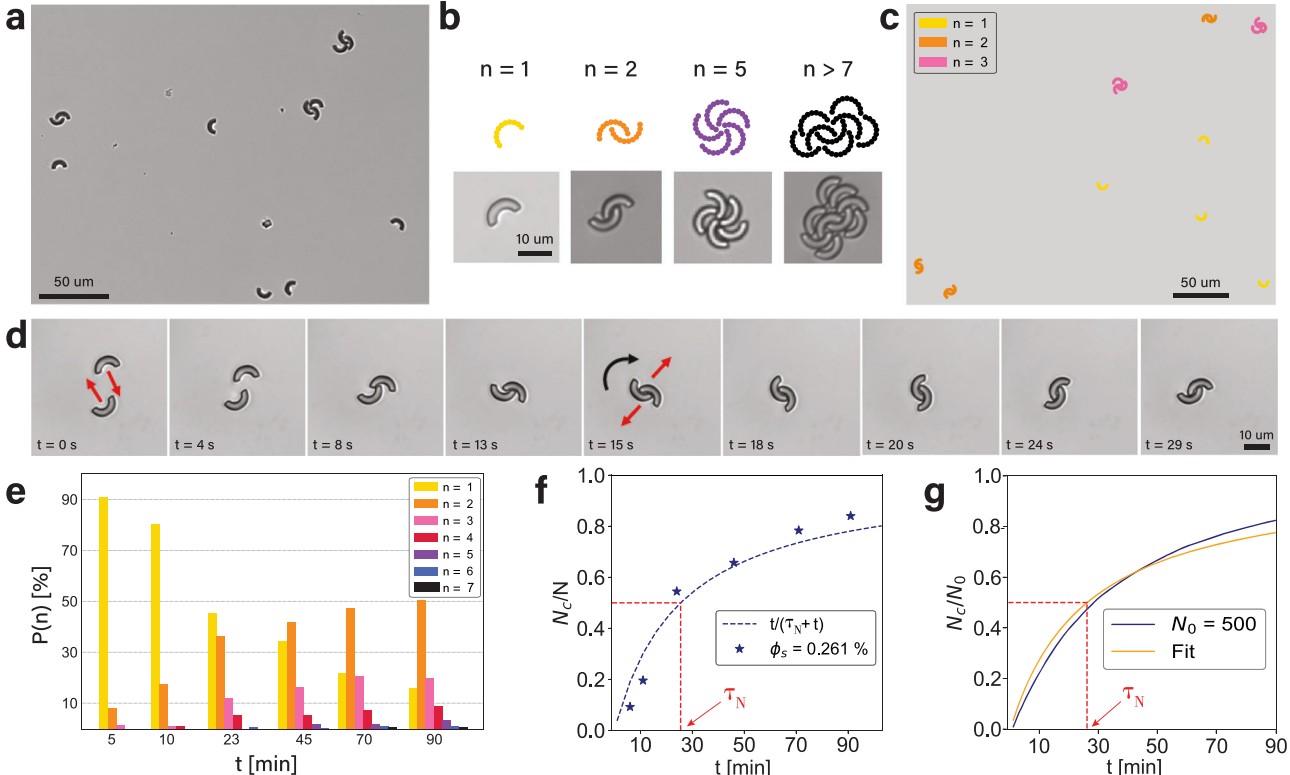

**Fig. 2 | Time-dependent clustering of active crescents. a** Bright-field microscopy image taken 90 min after mixing particles and fuel solution shows that interlocked pairs and clusters, as well as single particles are present (surface-area fraction $\phi_s = 0.261\%$, corresponding to 104 crescents/mm²). **b** Simulation snapshots and bright-field microscopy images of clusters of different size $N$. **c** Snapshot of simulations for $\phi_s = 0.261\%$. Clusters are colored according to size. **d** Image sequence of formation of a 180° crescent-pair in time. The pair only rotates after interlocking. **e** Time evolution of the probability $P(n)$ to find a crescent in a cluster of size $n$

($\phi_s = 0.261\%$, measured after 90 min). The average particle velocity does not change significantly even after long times (> 5h, see SI). **f** Time evolution of fraction of particles that are clustered, $N_c/N$, for the data shown in (**e**). Eq. (1) was used as a fit with $\tau_N = 25 \pm 4$ min. **g** Balls-into-bins model with $N_0 = 500$ and a system size comparable to experimental setup behaves similar to experiments and simulations. Fit of Eq. (1) with $\tau_N = 26.2 \pm 0.2$ min. Details in SI. Source data are provided as a Source Data file.

Not all collisions, however, will lead to the formation of clusters. The success depends on the probability for interlocking upon collision. For bend rods with an opening angle of $\alpha = 180°$ most collisions are successful: Besides direct interlocking upon collision, interlocking may occur after a short delay, i.e. when particles initially meet with their terminal parts and form a metastable ring-like structure. Any subsequent sliding induced by noise will result in interlocking, only with a delay. The quick formation and long stability of crescent-pairs seen in both experiments and simulations is very different from the behavior of pairs of active spheres, which tend to slide pass each other as soon as their velocities depart from a perfectly anti-aligned configuration. While cluster formation of spheres requires at least three, but typically more, particles[1,2,18], the crescent shape favors configurations in which the end of one crescent is locked at the center of another particle and thus already stabilizes clusters of two particles.

The stable, rotating pairs subsequently act as nucleation points for larger clusters. Notably, clusters only grow through addition of single crescents and not by cluster merging, because of a lack of translational motion once the crescents form a pair. The long lifetime of the pairs enhances nucleation, similar to active polygons studied in simulations[28]. But unlike active polygons, active crescents can interlock as their shape features a cavity. Interlocking has so far only been reported for passive colloidal particles[38,39] or active systems sometimes in combinations with passive agents[40–42], neither of which were, however, consisting of only self-propelled and anisotropic colloidal agents[40–42]. Here, the interlocking makes the pairwise self-assembly stable against break-up, even upon a collision with a free particle,

which is often absorbed into the cluster owing to its concave shape. We quantify the evolution of the cluster distribution over time and show the result for concave-side leading active bent rods (at $\alpha = 180°$, 1% $H_2O_2$) in Fig. 2e. Already after 10 min about 17% of crescents have formed pairs, despite our experiments taking place at a very low particle density of only $\phi_s = 0.261\%$, which corresponds to only about 104 particles/mm². Over the next 80 min the amount of single particles drops to 16%, more than half the particles are paired, while three- and four-particle clusters form in smaller amounts (20% and 9%, respectively).

This efficient nucleation and growth process leads to a quick assembly of the active bent rods into clusters. We find that the time evolution of the fraction of particles that are part of a cluster, $N_c$, relative to the total number of particles in the system, $N$, quickly grows such that the majority of particles are part of a cluster after only about 30 min, i.e. $N_c/N > 0.5$ (Fig. 2f), and almost all particles are part of a cluster after 90 min (Fig. 2e) despite the very low particle density of $\phi_s = 0.261\%$.

To better understand the observed clustering process we consider a balls-into-bins model, where particle clustering is modelled as the presence of multiple particles (balls) inside a finite set of containers (bins) after random assignment (see SI). For each time step, all single particles are randomly assigned a new bin. A $n$-particle cluster corresponds to a bin containing particles. Simulating many time steps, we find a $N_c/N_0$ curve as shown in Fig. 2g, which is qualitatively similar to those of the experiments.

To obtain an analytical expression, we consider only two-particle-cluster formation for simplicity, because dimers are

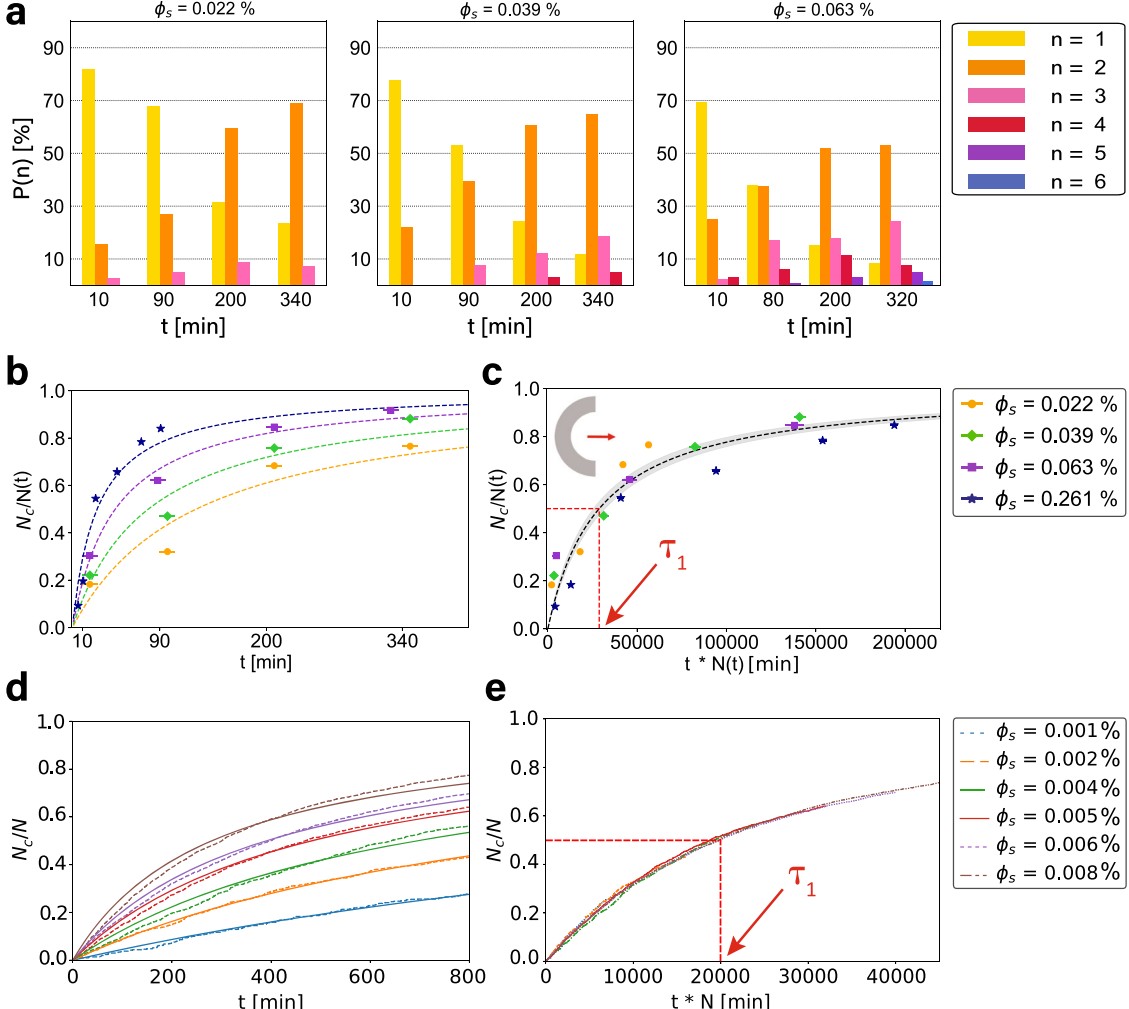

**Fig. 3 | Density dependent clustering of concave-side leading, bent rods** ($\alpha = 180°$, $L = 10\,\mu m$). **a** Cluster-size distribution over time for different surface-area fractions $\phi_s$, corresponding to particle number densities of (from left to right) 9, 15, and 25 crescents/mm². Note that $\phi_s$ is time-dependent and the stated values are measured after 90 min (see main text and SI). (b) $N_c/N(t)$ for different densities

fitted with Eq. (1). The observation time needed to cover the entire sample reflects as an $x$-error for each data point (also applicable to (**c**)). **c** Master curve obtained from (**b**) by rescaling time as $t \to tN(t)$. Fit with Eq. (1) with $\tau_1 = (2.9 \pm 0.3)10^4$ min. **d** $N_c/N(t)$ from simulations for various densities, and (**e**) the corresponding master curve, where $\tau_1 = (2.0 \pm 0.1)10^4$ min. Source data are provided as a Source Data file.

(initially) dominant. We then find that the time evolution of the average number of dimerized particles can be described by a Poisson process $\langle N_c(t) \rangle = 2rt$ with a time-dependent rate $r = r(t)$. Assuming a linear decrease of non-dimerized particles, $N_f = N - 2rt$, we find the Michaelis-Menten equation

$$\frac{\langle N_c \rangle}{N} = \frac{t}{\tau_N + t} \qquad (1)$$

where $\tau_N$ is the time required for two particles to collide and interlock (see SI). At short times $\langle N_c \rangle/N \sim t/\tau_N$ increases linearly with $t$, as the abundance of non-dimerized crescents makes the rate approximately time-independent. At long times, on the other hand, all particles are dimerized and $\langle N_c \rangle/N \to 1$. Fitting Eq. (1) to our experimental and numerical data we find very good agreement, as confirmed by Fig. 2f, g and Fig. 3b. In the following we will return to using the model of ref. 37 for simulations. The simulations presented are performed in the limit where translational and rotational noise are vanishing because when the rotational noise is small, the change of number of clustered particles is also relatively small. Only for large values the number decreases significantly, see SI for the effect of noise on the clustering dynamics.

## Density dependence of crescent clustering
Our initial results demonstrated that significant clustering already appears at very low particle densities. To investigate the density dependence of clustering and explore whether there is a limit, we tested three even lower particle concentrations in experiments. Remarkably, even at concentrations as low as nine crescents/mm² (i.e. $\phi_s = 0.022\%$) cluster formation still happens, see Fig. 3a, albeit slower. This surface area fraction is two orders of magnitude smaller than for active spheres[1,2,17], rods[29], and standing disks[32]. We return to simulations to investigate even lower particle densities. Similar to the experiments, a significant number of particles cluster (Fig. S11). Plotting $N_c/N$ in Fig. 3b, d, we find that the assembly occurs slower at lower particle densities, but still looks qualitatively similar. The Michaelis-Menten Eq. (1) fits all the simulation and experimental data well, independent of the density.

Eq. (1) suggests the presence of only a single time scale, the constant $\tau_N$, for cluster formation. This is surprising since the active Brownian motion of isolated particles, as well as the interaction between particles could, in principle, give rise to other time scales. To explain this we assume that particle collision occurs predominantly in the diffusive regime, where the particles' mean squared displacement is given by $\langle \Delta \mathbf{r}(t) \rangle^2 = 4\mathcal{D}t$, with $\mathcal{D}$ the diffusion coefficient. Thus, the

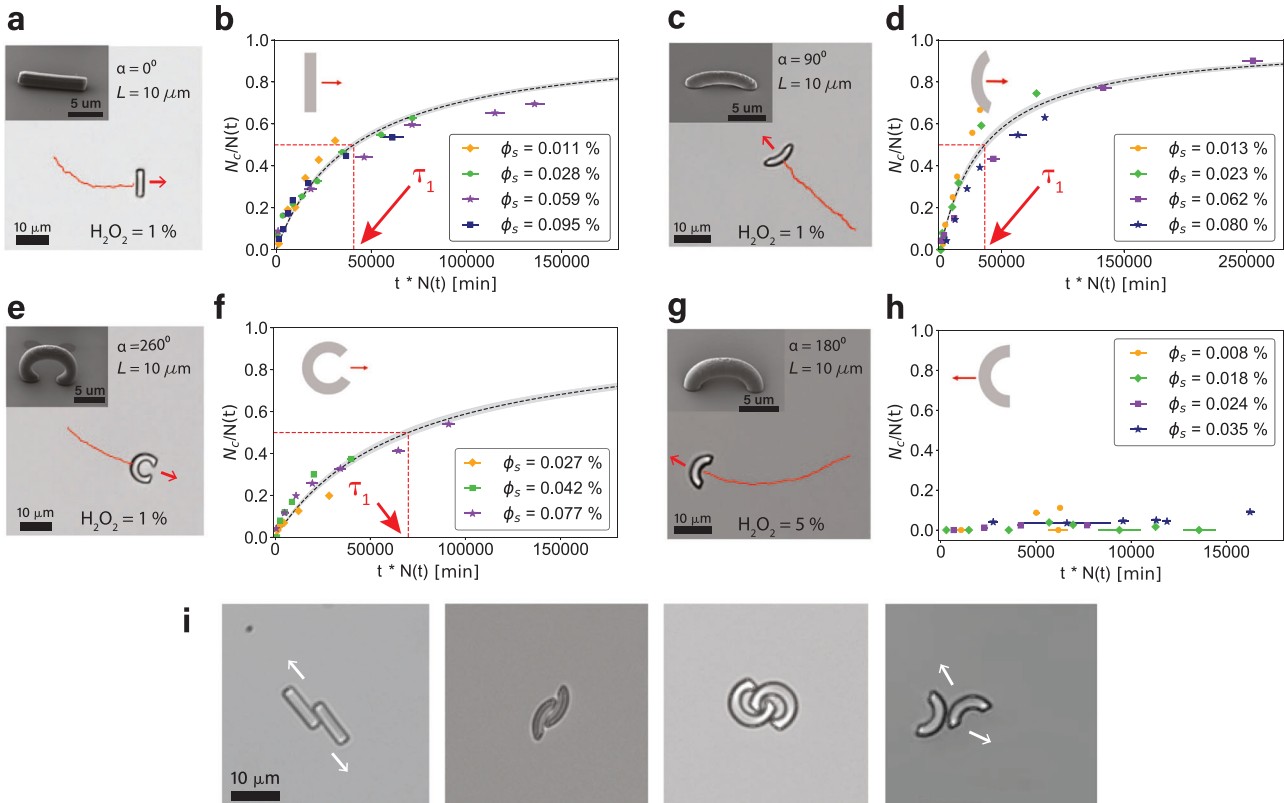

**Fig. 4 | Shape-dependent clustering behavior.** Scanning electron microscopy (inset) and bright-field images of 3D printed (**a**) straight rods ($\alpha = 0°$) and crescents with (**c**) $\alpha = 90°$, (**e**) $\alpha = 260°$ and (**g**) $\alpha = 180°$. The trajectory travelled by the active particle in the past 30 s is shown in red and the direction of motion is indicated by a red arrow. Fraction of particles in a cluster plotted after rescaling time as $t \rightarrow tN(t)$ for systems of (**b**) self-propelled straight rods, (**d**, **f**) concave-side leading crescents with (**d**) $\alpha = 90°$ and (**f**) $\alpha = 260°$, and (**h**) convex-side leading crescents with $\alpha = 180°$. The fit parameter obtained from fitting with Eq. (1) is (**b**) $\tau_1 = (4.1 \pm 0.2)10^4$

min, (**d**) $\tau_1 = (3.6 \pm 0.3)10^4$ min, and (**f**) $\tau_1 = (7.0 \pm 0.5)10^4$ min respectively. The $x$-error indicates the observation time needed to measure the entire sample. All surface area fractions $\phi_s$ correspond to particle densities below 50 crescents/mm². The exact values can be found in the SI. **i** Bright-field images of rods and crescent pairs that differ in their ability to interlock, cf. Supporting Video 7–12. The white arrow indicates how pairs that are not sufficiently stabilized by interlocking can slide past each other and break. Source data are provided as a Source Data file.

time for two particles to collide is approximately $\tau_N \sim \ell^2/(4\mathcal{D})$, with $\ell$ the average inter-particle distance. If particles are uniformly distributed in space, $(D/\ell)^2 \approx N$, with $D$ the system size, thus $\tau_N \approx D^2/(4\mathcal{D}N)$. Notice that, consistently with our combinatoric calculation, $\tau_N \sim 1/N$ (see SI). Transforming $t \rightarrow tN$ is then equivalent to rescaling time by the only time scale of the process, thus removing any particle-density dependence.

Indeed, with this simple rescaling of time, we obtain a master curve for the data from experiments and simulations described by a single fit parameter $\tau_1$, shown in Fig. 3c, e. We attribute the difference in fit parameter $\tau_1 = N\tau_N$ of approximately an order of magnitude to flow-induced attractions present in experiments as well as to slow sedimentation of particles which increases the surface concentration in time, see SI for a discussion. Both effects only modify the time scale but are not relevant for the basic clustering dynamics. We can thus conclude that the dimerization of active crescents is a single-time-scale process, regardless of the particle density. Furthermore, clustering occurs even at arbitrarily low concentrations. This is in stark contrast to spheres, which undergo a dynamic clustering process that consists of break up and growth, and the transient pairs formed by rods and disks. This peculiarity of active crescents ultimately determines the high performance of their assembly and originates from the efficiency of the interlocking mechanism governing dimerization. This implies an infinite interaction time and, effectively, a vanishing ballistic-diffusive crossover time thus leaving $\tau_N$ as the only finite time scale of the process.

## Shape dependence of crescent clustering

To study the connection between shape and collective behavior we print two other types of bent rods, one with a smaller opening angle $\alpha = 90°$ and one with a larger opening angle $\alpha = 260°$, keeping $L$ constant, see Fig. 4c and e. We measure their clustering behavior for different surface area fractions and, strikingly, find in both cases that the $N_c/N$ data can again be rescaled into a single master curve. The fit parameter $\tau_1$ increases from $\tau_1 = (2.9 \pm 0.3)10^4$ min for $\alpha = 180°$ (Fig 3c) to $\tau_1 = (3.6 \pm 0.3)10^4$ min for $\alpha = 90°$ (Fig. 4d) and $\tau_1 = (7.0 \pm 0.5)10^4$ min for $\alpha = 260°$ (Fig. 4f), reflecting the slower assembly process.

To better compare these numbers, we also need to take into account the particles' speed, as a higher speed will lead to faster assembly. At 1% $H_2O_2$ the speeds for 90° crescents at $\langle v \rangle_{90} = 1.02 \pm 0.03$ µm s⁻¹ were comparable to that of the 260° crescents, $\langle v \rangle_{260} = 1.09 \pm 0.01$ µm s⁻¹, implying that the 260° crescents assemble significantly slower. However, in both the 90° crescents and the 260° crescent case, the speed is higher than the speed of the 180° particles, $\langle v \rangle_{180} = 0.78 \pm 0.08$ µm s⁻¹ making the difference in the characteristic assembly time $\tau_1$ even more noteworthy. Moreover, the number density of particles in the sample is higher at the same $\phi_s$ for the 90° crescents, due to their shorter arc-length, underlining the difference further.

To investigate if interlocking due to the concave shape is important, we execute two control experiments. First, we print straight rods, i.e. $\alpha = 0°$, and quantify their clustering behavior in time for different

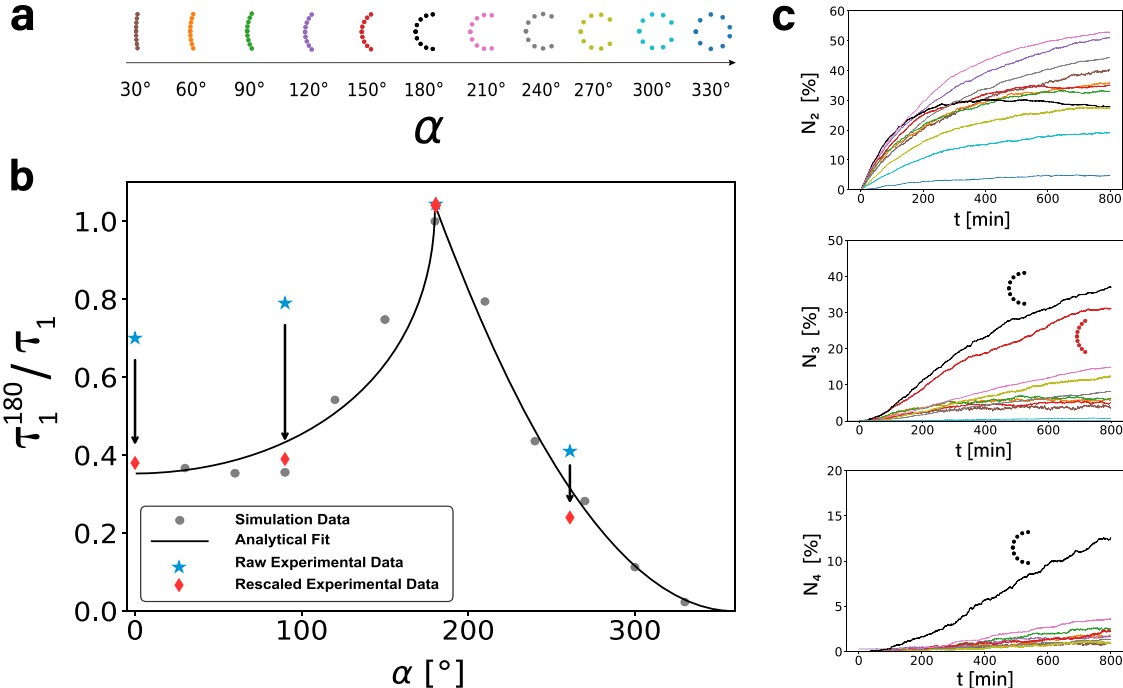

**Fig. 5 | Simulation results on the influence of the opening angle $\alpha$ on the clustering behaviour. a** Bent rods with increasing opening angle from 30° to 330° used in the simulations shown in (**b**). **b** Inverse fit parameter $1/\tau_1$ as a function of the opening angle $\alpha$, normalized to the value of $\tau_1$ for $\alpha = 180°$. The fit is found from a simple model describing cluster creation and decay, see the main text for details. The fit shown here is $\tau_1^{180}/\tau_1 = 2.357(1.149 - 1/\sqrt{d(\alpha)^2 + 1})\Theta(\pi - \alpha) + 0.106 P_{cl}\Theta(\alpha - \pi)$,

with $\Theta$ the Heaviside step function. The red data points (star) represents the normalised inverse fit parameter for straight rods, 90° crescents and 260° crescents (from left to right) found in experiments before rescaling, the blue data points (diamond) the value after rescaling (see main text and SI). **c** Evolution over time of the number of particles in a two-particle-cluster (top), a three-particle-cluster (center) and a four-particle-cluster (bottom), respectively. Source data are provided as a Source Data file.

surface area fractions, see Fig. 4a, b. Straight rods can form pairs too[29] but can still slide past each other leading to a breakup of the cluster Fig. 4i and Supporting Video 9. At 1% $H_2O_2$, the rods move with an average speed of $\langle v \rangle_0 = 1.58 \pm 0.03 \ \mu m \ s^{-1}$ and, surprisingly, follow the same clustering dynamics as the bent rods with a fit parameter for the master curve $\tau_1 = (4.1 \pm 0.2)10^4$ min. This means that the clustering behaviour of active straight rods is within error comparable to the one of 90° bent rods, noting that the speed of the straight rods was higher than of any of the crescents.

The second control group are spherical particles, which are at the other end of the shape spectrum. This shape change can be effectively implemented by crescent particles that move in the direction of and interact with their convex side instead of their concave side, see Fig. 4g, h, with effectively a collision interaction comparable to those of spheres in 2D experiments. We purposefully did not print bent rods upside down to obtain convex-side leading swimmers because such particles would show a truncated shape with a flat surface at their convex side due to the contact area between printed structure and substrate required in the printing process. This flat surface leads to a significantly higher stability of pairs and thus affects the clustering dynamics. Instead, we experimentally achieve convex-side leading crescents by increasing the $H_2O_2$ concentration to 5%, which leads to a direction reversal, and term these $c^+$ crescents, following ref. 27. While their average velocity at $1.3 \pm 0.2 \ \mu m \ s^{-1}$ was slightly higher than that of the concave-side-leading particles, their efficiency to cluster at low density drops significantly (Fig. 4h). The few pairs observed over the course of the experiment typically broke up by sliding past each other within ~ 2 min (Fig. 4i and Supplementary Video 12). Fig. 4i shows pairs of straight and bend rods of different opening angles and which all differ in their ability to interlock (also see Supplementary Video 7–11). For all particle shapes that significantly slow down the motion, i.e. straight rods and concave-side leading bent rods, we

observe clustering despite the low surface densities. This motility induced clustering results from a combination of the break down of detailed balance and steric interactions. In bulk systems, a similar mechanism drives phase-separation at small length scales (MIPS). Here we showed that the same mechanism can lead to a cluster phase even in the absence of a true bulk phase, and hence term it motility induced clustering.

To study many different opening angles we turn again to simulations. We consider eleven different opening angles from 30 ° to 330° in steps of $\Delta\alpha = 30°$, keeping the cross-section fixed, see Fig. 5a. For fixed arc-length and varying cross-section, the particles cluster at a similar speed, see SI. When again plotting $N_c/N$ we find that 180° crescents cluster most efficiently (cf. Fig. 5b) with the smallest value for $\tau_1$. To quantify this behavior we present in Fig. 5c the inverse fit parameter $1/\tau_1$ as a function of $\alpha$, normalized to the value found for $\alpha = 180°$. The larger $1/\tau_1$, the faster the clustering. We find the largest value for 180° crescents, with $1/\tau_1$ being approximately symmetric about $\alpha = 180°$ up to 90°. However, increasing the opening angle further, $\tau_1$ increases sharply, while with decreasing opening angle the fit parameter reaches a plateau around $\alpha = 90°$.

To rationalize the dependence of $\tau_1$ on $\alpha$ we assume two competing effects are relevant. This approach is similar to ref. 43 where, however, a three-dimensional system of passive bent rods was considered which results in significantly different dynamics. 1) the probability $P_{cl}$ to form a cluster and 2) the stability of the cluster. $P_{cl}$ depends on the opening length $L_{op}$, which is independent of $\alpha$ for $\alpha < 180°$. However, for $\alpha > 180°$, $L_{op}$ and thus $P_{cl}$, decrease with increasing $\alpha$. In the extreme example of a sphere, stable dimers cannot form. The probability of two particles colliding at an angle such that their cavities are facing each other, resulting in a dimer, is $P_{cl} \sim (\alpha - 2\pi)^2 \sim 1/\tau_1$.

However, once a cluster has formed, its stability depends on $\alpha$, too. For smaller $\alpha$, particles interlock less and cluster are less stable,

e.g., in case of a collision with a free particle. As a simple proxy for stability we define the length $d(\alpha) = \sqrt{(L_{op}/2)^2 + h^2}$ that one crescent in a dimer would have to move to leave the cluster. The smaller this length, the more unstable the cluster. This length scale yields a time scale $\mathcal{T} \sim d/v_0$ such that $\tau_1 \sim \mathcal{T}$ captures the $\alpha$-dependence of the parameter $\tau_1$. We find $1/\tau_1 \cdot 1/d(\alpha)$. We find that combining these two effects results in a good agreement, see Fig. 5c, especially considering the simplified arguments we used.

Cluster formation is optimal for 180° crescents because the probability for cluster formation is high, due to their relatively large opening length, and, once formed, clusters exhibit significant stability from the particles' strong curvature. In experiments we also find $1/\tau_1$ to be highest for 180° particles. However, the values found for straight rods, 90° crescents and 260° crescents are significantly higher than the values found in simulations, see blue data points in Fig. 5b. We can rationalize this difference from the different propulsion speeds and sedimentation rates of those particles. By rescaling the inverse and normalized fit parameters $\tau_1^{180}/\tau_1$ with the measured propulsion velocities for the respective shapes as well as with the corresponding inverse calculated sedimentation velocities (see SI and[44]), we obtain corrected values for $\tau_1^{180}/\tau_1$ which are in perfect agreement with the values found in simulations, implying again that hydrodynamic interactions are negligible, see red diamonds in Fig. 5b.

As a consequence of the shape-dependent competition between interlocking and breaking-up, higher-order cluster are also more stable for 180°. Three-particle-clusters are already most prevalent for 180° particles, but four-particle-clusters are effectively only observed for $\alpha = 180°$ (see Fig. 5c). While the absence of higher order clusters in samples with higher opening angles can be explained by the increased difficulty to form these types of structures, for lower opening angles, the lack of larger clusters arises from the instability of two- and three-particle clusters when colliding with free particles.

## Discussion

In this manuscript we investigated the clustering behavior of self-propelled particles with a shape that is continuously tunable along a single anisotropy dimension by exploiting a combination of 3D printing and simulations. Printing bent rods with different opening angles allowed connecting our results with the two major shapes thus far employed, spheres and rods. Surprisingly, stable interlocking into pairs occurs at arbitrarily low number concentrations of particles due to a interlocking type interaction. Pair formation plays a critical role in nucleation, and the dynamics of the fraction of clustered particles can thus be captured with a notably simple formula which allows rescaling of the clustering dynamics into a single master curve. We find that particle shape influences both the assembly speed and cluster size distribution with an optimum for crescents with an 180° opening angle and can be described by a competition between interlocking and break-up.

Our experimental and numerical results provide the first detailed understanding on how a gradual shape change from a sphere to a rod affects the clustering behavior of active particles and uncover a simple yet general description for this complex effect. These key insights are important to understand how non-spherical shapes affect biological swimmers, and can be leveraged to further design interlocking type connections between active particles to control the stability of complex self-assembling structures through the depth of their entangling sites independent of their composition and surface chemistry. This strategy offers the unprecedented opportunity to design hierarchical assembly pathways towards active functional materials through engineering the active particle shape. The guaranteed assembly even at extremely low concentrations together with the interlocking shape recognition furthermore opens up possibilities in the drug delivery where site recognition or a low concentration of the drug-carriers as

well as the activation of reactants after assembly is important. In the future, it will be interesting to study the cross-over from our low-density system which exhibits motility induced clustering of mostly small clusters to a system with higher surface area density where larger clusters with different assembly dynamics and motility induced phase separation (MIPS) may occur.

## Methods

### Reagents
Fused silica substrates as well as the photoresist IP-Dip were acquired from Nanoscribe GmbH. Propylene glycol methylether acrylate (PGMEA, ReagentPlus ≥99.5%) was purchased from Sigma Aldrich and isopropanol (IPA) was obtained from VWR chemicals. Hydrogen peroxide ($H_2O_2$, 35 wt% solution in water, stabilized) was purchased from Acros Organics. Water was purified by means of a MilliQ system (resistivity ≥ 18 MΩ.cm). Unless otherwise noted, all chemicals were of analytical or reagent grade purity and were used as received from commercial sources.

### Particle fabrication
All particles were 3D printed on a Nanoscribe Photonic Professional GT microprinter equipped with a 63x oil-immersion objective (Zeiss, NA = 1.4) following[35]. Particles were printed on a fused-silica substrate using IP-Dip as photoresist. Bent rods with an opening angle of 180° and 260° were designed with a thickness of 2 μm and a cross-section length $L$ of 10 μm. For bent rods with an opening angle of 90° the thickness of the particles was reduced to 1.5 μm in order to keep the same size ratio between the thickness of the crescent legs and the concave opening. Straight rods were designed with a thickness of 2 μm and a length $L$ of 10 μm. During development the printed particles were submerged in PGMEA for 30 min followed by 2 min in IPA and then left to dry overnight at ambient conditions. Once dry, active crescents were coated with a 5 nm layer of catalyst (Pt/Pd 80:20) using a Cressington 208HR sputter-coater. To prevent excess Pt/Pd, the area around the print was protected with tape which could be removed later without damaging the printed structure. After sputter coating, each print was checked by default under the microscope to determine its quality. The substrate was placed onto a 200 μL MilliQ water droplet in the center of a small glass petri dish such that the printed crescents were in contact with the water. Repeated sonication for 2 min allowed recovery of 90% of the particles from the substrate. Particles were subsequently concentrated by centrifugation followed by removal of the supernatant.

### Imaging, data acquisition, and data analysis
For observation of the clustering dynamics, a Nikon Eclipse Ti-E bright-field light microscope with a Plan Apo λ 20x long working distance objective (NA = 0.75) was used. Pt-coated colloidal particles were suspended in freshly prepared 1% hydrogen peroxide aqueous solution in the case of concave-side-leading crescents and rods, or in 5% hydrogen peroxide for convex-side-leading crescents. Control experiments with Brownian crescents were performed in MilliQ water. This colloidal solution was then placed into a sample holder (∅ = 8 mm) using an untreated borosilicate glass coverslip (VWR, 25 mm, No. 1) as substrate.

The cluster distribution in the sample was recorded after different time steps. We hereby defined a cluster as an assembly of at least two particles that remained stable for at least 5s. A square array of 7 × 7 images (2048 × 2048 px) with an overlap of 1% was taken to cover and observation area of 4.5 mm x 4.5 mm in the central area of the sample holder. The time needed to take these 49 single images including corresponding 5s videos for each field of view resulted in an average value and error for the determination of the time-point. Short videos were only taken for samples with lower particle densities where the total observation time exceeded 100 min. All

measurements were taken in the dark and in the absence of bubble formation.

Single crescents as well as crescent cluster species were counted for every time step, where single stuck particles (as determined from 5s videos) were left out of the count of single crescents. Average particle velocities were captured from 30s long videos with a frame rate of 20 fps. Particles were tracked in each frame, applying the Canny Edge detection algorithm to generate a mask out of which the particle center of mass was obtained. Individual crescent-velocities $v$ were determined by fitting the short-time regime ($\Delta t \ll \tau_r$) of the mean squared displacement (MSD) of different crescents with $\Delta r^2 = 4\mathcal{D}\Delta t + v^2\Delta t^2$. The second fit parameter is the diffusion coefficient $\mathcal{D}$ and $\tau_r$ was chosen as the rotational diffusion time for a sphere. We here use the aforementioned fit-equation, originally developed for spherical active particles[36,45] as an approximation for our crescent-shaped particles despite them not being spherical.

## Dynamic simulations

Following Wensink et al.[37] we simulate $N$ self-propelling particles in two dimensions that move with a velocity $v_0$. Each particle is discretized into $k$ equidistant spherical segments of diameter $d$ (see Supplemental Figure 2). A particle $\rho$ has an orientation $\boldsymbol{u}_\rho$ and position $\boldsymbol{r}_\rho$. The position of segment $i$ with respect to the position of the center of mass $\boldsymbol{r}_\rho$ is denoted by $\boldsymbol{e}_\rho^i$. The pair potential of two particles $\rho$ and $\delta$ is given by $U_{\rho\delta} = k^{-2} \sum_{i,j=1}^{k} u(r_{\rho\delta}^{ij}/d)$. Here, $u(x) = u_0 \exp(-x)/x^2$ is a short-range potential that is repulsive if $u_0 > 0$ resulting in effectively hard particles. $r_{\rho\delta}^{ij} = |\boldsymbol{r}_\rho - \boldsymbol{r}_\delta + \boldsymbol{e}_\rho^i - \boldsymbol{e}_\delta^j|$ is the distance of two segments of the two different particles. The equations of motion, found from balancing forces and torques due to activity and steric interactions, for a particle $\rho$ are then given by[37]:

$$f_t \partial_t \boldsymbol{r}_\rho = F_\text{a} \boldsymbol{u}_\rho - \boldsymbol{\nabla}_{\boldsymbol{r}_\rho} U + \xi \tag{2a}$$

$$f_r \partial_t \varphi_\rho = -\boldsymbol{\nabla}_{\varphi_\rho} U + \eta \tag{2b}$$

where $f_t$ and $f_r$ are translational and rotational friction, respectively, $\boldsymbol{u}_\rho = \{\sin\varphi_\rho, \cos\varphi_\rho\}$, $v_0 = F_\text{a}/f_t$, and $U = \sum_{\rho,\delta(\rho\neq\delta)} U_{\rho\delta}/2$. $\xi$ and $\eta$ are translational and rotational Brownian noise, respectively.

We simulate the particles in a square system with periodic boundary conditions. A number of $N$ particles is initialized in a system of size $D \times D$ with random orientation and position. In the simulations we have defined a particle to be in a cluster with another particle if it is closer than $3R$ for 100 iterations. Unless otherwise noted, each of the results presented is found by averaging over 100 independent runs. We employed the same particle dimensions and velocity in the simulations as in experiments. A list of parameters used, the mapping between computational and physical units, as well as additional details can be found in the Supplementary Methods section.

## Balls-into-bins model

The dynamics of $N$ active particles of cross-sectional length $L$ moving with speed $v_0$ in a square system of width $D$ are mapped onto a balls-into-bins model. For this, we choose the number of boxes to be $B = (D/S)^2$, with $S \approx L$ the size of a single box. In the first step of the simulation we randomly assign each of the $N$ balls a box. If $n > 1$ balls are assigned to the same box, we count this as a $n$-cluster. If a ball is in a box by itself, it is counted as a free particle. In the following step of the simulation the above procedure is repeated, but only *free* particles are assigned a new box while $n$-clusters remain in their respective box. See the Supplementary Methods for a more detailed explanation, a list of parameters used, and a mapping between computational and physical units.

## Data availability

Source data are provided with this paper and can be found through the 4TU public data repository under the link: https://data.4tu.nl/datasets/eede1636-db08-4b0f-b270-20c368031c50    https://doi.org/10.4121/eede1636-db08-4b0f-b270-20c368031c50.

## Code availability

Simulation codes are available under the following link: https://github.com/ludwighoffmann/CAPI.git.    https://doi.org/10.5281/zenodo.11504391.

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

## Acknowledgements
This work was supported by the Netherlands Organization for Scientific Research (NWO/OCW), as part of the Vidi scheme (VI.Vidi.193.069, D.J.K. and 680-47-547, L.G.). We thank Rachel Doherty for support with 3D printing. Part of this work was performed using the ALICE compute resources provided by Leiden University.

## Author contributions
D.J.K. and S.R. conceived the work. S.R. carried out the experiments. L.A.H. and L.G. developed the theory, and L.A.H. performed the simulations. D.J.K. supervised the work. All authors discussed and interpreted the results and contributed to writing the final manuscript.

## Competing interests
The authors declare no competing interests.
