## [Peer Review File · Nature Communications]

REVIEWER COMMENTS

Reviewer #1 (Remarks to the Author):

This manuscript by Reidel et al. demonstrates the formation of clusters in low densities of bent-rod shaped active particles. The authors use two-photon polymerization-based 3D printing to fabricate the particles that are partially coated with Pt/Pd and dispersed in H₂O₂, where they are self-propelled – presumably due to a self-phoresis mechanism. Through a combination of shape-anisotropy, self-propulsion, and steric hindrance, they show the formation and growth of clusters of different sizes over time. The cluster growth rate is well described by a scaling parameter determined only by particle density, indicating the minimal influence of hydrodynamic and phoretic interactions in the system. The authors further study the effect of particle shape by using colloids with a lower opening angle and observe slower growth dynamics owing to the lower stability of particle clusters. Similarly, in a system of particles propelled towards their concave-side, the authors observe no cluster formation as this propulsion mode does not allow for particle inter-locking. The experiments are accompanied by some elegant, if simple, simulations and modelling where the authors are able to obtain a master curve depending only on particle density and shape, that captures the experimental data quite well, and indicates the optimal particle geometry for cluster formation ($\alpha = \pi$). The simulation results are extensive and cover a number of relevant parameters.

One of my main issues is that the authors seem to overstate the novelty and advances made in this work. The basic idea of self propulsion+interlocking shapes leading to cluster formation is somewhat obvious, and geometrical interlocking/surface interactions of active particles with other active/passive structures has been described before (Shields IV, C. W., et al. *Advanced Functional Materials*, 28(35), 1803465., Maggi, C. et al., *Small*, 12(4), 446-451, Gao, W. et al. *Journal of the American Chemical Society*, 135(3), 998-1001). The authors also repeatedly refer to a lock-and-key mechanism, which I do not believe is appropriate for this experimental system. A lock-and-key system implies the existence of multiple species (at least two) with unique pairing mechanisms existing between two complementary species. I certainly would like the authors to consider providing better context for this work.

Overall, the experiments in this manuscript are well presented and supported by illuminating simulations and a simple model. This work is generally suitable for publication, but I am not convinced that it advances the field sufficiently in its current state to merit publication in *Nature Communications*.

Below are my comments on the manuscript:

1. The authors show experiments of 2 different geometries ($\alpha = \pi, \pi/2$) which seems limited considering the broad conclusion regarding the derived master curve and the claim of ‘continuous shape tunability’. Experiments with at least one $\alpha > \pi$ condition might be useful since additional symmetry effects will come into play both in the propulsion mechanism of the particles and particle/particle interactions.

2. The reversal in swimming direction at 5% H₂O₂ is surprising. Can the authors provide a mechanism that leads to this reversal in swimming direction? Also it seems simpler to print the rods in the opposite direction and coat the concave side to obtain convex-side leading swimmers. This would avoid potential changes to inter-particle interactions arising from using high [H₂O₂].

3. It might be useful to measure the lifetimes of different clusters sizes and discuss the dynamics observed in high n clusters. It is possible that at high n, the clusters are only metastable due to crowding effects and phoretic interactions will likely play a larger role here.

4. The authors report experiments over remarkably long periods (90 mins) for this system. Do the authors notice any changes to the observed particle velocities due to fuel depletion effects – especially at higher particle concentrations.

Minor comments:

1. Figure 3 caption contains panel (f) which seems to be missing in the figure.

2. I also did not find any SI videos connected to Fig4, showing clustering dynamics in $\alpha = \pi/2$ swimmers.

Reviewer #2 (Remarks to the Author):

In the manuscript “Designing highly efficient lock-and-key interactions in anisotropic active particles”, the authors report the cluster formation by anisotropic swimmers and systematically investigate experimentally and by simulations the dynamics of clustering as a function of shape at extremely low densities. The authors demonstrated that the stability of clustering (rather interlocking) is strongly influenced by the shape of the swimmers. The analytical model shows that the dynamics of clustering can be captured by a Michaelis-Menten kinetics with a single scaling parameter.

Overall, the paper is clearly written and touches upon a rapidly developing area in the soft matter community addressing the effects of shape on collective dynamics in active ensembles. It definitely will be of interest to the active matter community and, in my opinion, is appropriate for Nature Communications.

I have a few minor comments for the authors.

- I think, the lock-and-key analogy is not appropriate for this particular system as both particles involved are identical (in contrast to the original paper from the David Pine).

- The authors in their abstract and in the conclusion state that this work opens up possibilities for “designing targeted two-component drug delivery” that is not demonstrated in the paper at any level and just sounds a bit bombastic.

- The introduction in general does a good job overviewing dynamics in active systems formed by shape anisotropic particles. I agree with the authors' statement that there are not many such systems realized experimentally. There is, nevertheless, the recent active system formed by shape-anisotropic (pear-shaped) Quincke rollers that also demonstrate very unusual clustering in response to the activity changes (Nature Communications 11, Article number: 4401 (2020)). Authors might find it useful for the introduction.

Reviewer #3 (Remarks to the Author):

The authors present an experimental and numerical study of the clustering behavior of active bent rods. The bent rods are microswimmers that self-propel, whose shape can also be tuned. The interlocking of the rods can facilitate cluster formation, allowing clustering even at very low particle densities. Different shapes were investigated and semicircles were found to be most effective for particle clustering. The clustering dynamics was explained by a single master curve.

After reading the manuscript, I have two major concerns. First, the authors borrowed the concept of lock and key and use it in active system. This is appealing, but the system presented is about the interlock of two bent rods. I find difficult to relate it to the lock and key model, where shape complementary helps with interactions. In the current system, what is the lock and what is the key? Because the title of the manuscript is "Designing highly efficient lock-and-key interaction", some introduction and proper reasoning about the lock-and-key interaction is needed. Regardless, I feel it is just the "interlock" of two convex particles rather than "lock and key".

The second concern is about the conclusion that clustering can occur at low particle density, which the authors tried hard to sell. Given the fact that the particle can interlock due to their convex shape, is it obvious that they can form pair at low concentration (as long as particle can meet in the right orientation)? Once the pair forms, it seems to be irreversible (due to the propulsion). The author compares their system with mobility induced phase separation (MIPS) and few other systems, is it a fair comparison? Of course it is the authors' credit to report a system that is different, yet I feel the conclusion is somewhat trivial, although the anisotropic shapes indeed plays its role in forming and stabilizing the cluster. I will need to leave this point for other reviewers to judge.

I have some other comments for the reviewers to consider:

1. When tuning the shape, straight rod is important as a control experiment. Can they also form pairs. The authors should add experiments and analysis about it. In this case, no interlock is available and can be compared properly with one with interlock.

2. Why do the bent rods move in opposite directions with 1% H₂O₂ and 5% H₂O₂?

3. Have the author studied particle with a very high density, to study how the interlock influence the cluster formation and dynamics. As the cluster grow larger, it seems that the kind of interlocking for pairs has changed to other configurations and allows more freedom to move around. The so-called “lock and key” interaction has been weakened.

4. While such shape-induced clustering indeed inspire readers, several key points are less discussed: (i) mechanism of the active lock-and-key interaction. The author may provide force analysis on the dimer and trimer in the Video S5. As not all two-particle collision results in formation of the dimer. What kind of kinetic pathway is required. (ii) shape/activity-dependent structures. In current dimers (Figure 2b), the terminal of one particle locates at the center of another particle. If speed or particle shape (opening angle) change, whether the structure changes or even become unstable? These factors may affect the assembly efficiency.

5. Brownian motion is negligible for the 10- μ m particles, which may benefits the stabilization of clusters. In the presented simulation, noises are ignored with vague description “the noise did not change the behavior significantly.” While extensive biological and synthetic swimmers could be smaller and bear the influence of the Brownian motion. Can the author analyze the influence of the particle size on the “active lock-and-key interaction” by the experiment or the simulation? Admittedly, synthesis of the smaller concave particles could be difficult.

Point by point reply

(Dated: April 30, 2024)

I. REPORT OF THE FIRST REFEREE

One of my main issues is that the authors seem to overstate the novelty and advances made in this work. The basic idea of self propulsion+interlocking shapes leading to cluster formation is somewhat obvious, and geometrical interlocking/surface interactions of active particles with other active/passive structures has been described before (Shields IV, C. W., et al. *Advanced Functional Materials*, 28(35), 1803465., Maggi, C. et al., *Small*, 12(4), 446-451, Gao, W. et al. *Journal of the American Chemical Society*, 135(3), 998-1001).

- We thank the referee for taking the time reviewing our manuscript and for their suggestions for improvement. However, we respectfully but strongly disagree with the statement that our work may be considered limited in its novelty, and that a phenomenon, that to our eye, is far from easily predictable is termed "obvious". Even if the two-particle interaction might seem straightforward, the subtlety of the clustering mechanism unraveled in this work and the meticulousness of our analysis, which combines experiments and multiple theoretical approaches is not. In fact, as soon as one thinks about the problem the apparent simplicity disappears: our bent-rod colloids are subject to forces - e.g. Brownian, steric and viscous - which are expected to rip the clusters apart. Yet, they persistently cluster, do so much more efficiently than spheres, and all follow the same statistics (see e.g. Fig. 4 in the manuscript). Active particles with this shape have not been experimentally explored before, and this is the first experimental demonstration that shape can enhance clustering in systems of active particles. In addition, the precise fashion how the shape and hence interlocking translates into collective behavior is not trivial, and we describe it for a whole class of shapes. We consider neither the statistics (Michaels-Menten etc.) and efficiency of clustering, nor the density-dependency as obvious. Moreover, we find the simple dependence of the clustering efficiency on shape and the identification of an optimum surprising and exciting. As such our work is the first of its kind and hence clearly novel.

Our work is also distinct from the articles referred to by the referee. While these works show some form of interlocking, none of them studies the interactions and collective behavior of all self-propelled, anisotropic colloidal particles. Instead, they consider how spherical active colloids get trapped into cavities of a passive object (Maggi, C. et al., *Small*, 12(4), 446-451), or study the interactions between significantly larger rotors which are neither thermal nor show translational self-propulsion, but use external fields and only for inducing rotation (Shields IV, C. W., et al. *Advanced Functional Materials*, 28(35), 1803465), and finally, they consider the self-assembly of isotropic Janus micromotors into anisotropic structures (Gao, W. et al. *Journal of the American Chemical Society*, 135(3), 998-1001). This is clearly different from investigating the interactions of all self-propelled and anisotropic microscopic agents which we consider here, even though some form of interlocking is present in all of them. We now refer to these earlier works when discussing the pair formation to clarify the commonality of interlocking and the differences and advances of your work better.

- **Changes to the manuscript:** We added references to our manuscript to clarify that interlocking is a mechanism that occurs in other systems as well, see section *Collective Behavior of Active Bent Rods*, paragraph 4, page 4.

The authors also repeatedly refer to a lock-and-key mechanism, which I do not believe is appropriate for this experimental system. A lock-and-key system implies the existence of multiple species (at least two) with unique pairing mechanisms existing between two complementary species. I certainly would like the authors to consider providing better context for this work.

- We understand the referee’s concern regarding the use of the term lock-and-key mechanism. Our intention in choosing this analogy was to emphasize how strikingly well the interlocking works, most importantly of course in the case of bent rods with an opening angle of π . To eliminate any ambiguity, we revised the text and exchanged the term lock-and-key by the term interlocking.
- Changes to the manuscript: The term lock-and-key was replaced by the term interlocking.

Major points:

1. The authors show experiments of 2 different geometries ($\alpha = \pi, \pi/2$) which seems limited considering the broad conclusion regarding the derived master curve and the claim of ‘continuous shape tunability’. Experiments with at least one $\alpha > \pi$ condition might be useful since additional symmetry effects will come into play both in the propulsion mechanism of the particles and particle/particle interactions.
 - Our original choice to experimentally show three geometries was made because of the time needed to do and analyse the experiments. However, we agree with the referee that our manuscript benefits from additional experiments and have therefore taken additional data for bent rods with an opening angle of 260° as requested. In addition, we performed a series of control experiments with active rods ($\alpha = 0$) and hope that with these two new shapes it will become even more clear how well our experimental data matches the data obtained from simulations.
 - Changes to the manuscript: The manuscript was modified to include the experimental results for rods ($\alpha = 0$) and crescents with an opening angle of 260° , see section *Shape Dependence of Crescent Clustering*, paragraph 1, page 5-7 as well as Fig 4 and 5 and the SI.
2. The reversal in swimming direction at 5% H₂O₂ is surprising. Can the authors provide a mechanism that leads to this reversal in swimming direction? Also it seems simpler to print the rods in the opposite direction and coat the concave side to obtain convex-side leading swimmers. This would avoid potential changes to inter-particle interactions arising from using high [H₂O₂].
 - We thank the referee for raising this point. Indeed, the mechanism for direction reversal is at this point unknown and we are currently doing further experiments to pinpoint the origin of this behavior. Just like the referee suggests, we had initially planned to print the rods in the opposite direction and coat the concave side to obtain convex-side leading swimmers. However, after printing and inspecting the particles we decided against using those particles because 3D printing requires a non-zero contact area between the particle and the substrate to ensure good attachment. After development these particles therefore show a truncated shape with a flat surface at their convex side as can be seen in the below image. This flat surface leads to a significantly higher stability of pairs of bent rods and thus affects the clustering dynamics. This flat surface is not present for the bent rods swimming in the reverse direction and hence we chose for those.

We agree that other inter-particle interactions might arise from using a different fuel concentration. However, as we have already shown in the manuscript, flow-induced attractions for concave-side leading crescents in low fuel concentrations only have an impact on the time scale of our experiments but do not modify the basic clustering dynamics. We expect that the same reasoning also applies to convex-side leading crescents in higher fuel concentrations and this is supported by our simulation results. Even more so, as reported we

find that convex-side leading crescents in 5% H_2O_2 have a slightly higher average velocity than concave-side leading crescents in 1% H_2O_2 . This in principle should favour clustering but still no clustering is observed. These findings led us to believe that the clustering behavior of convex-side leading crescents is dominated by shape effects. An example of the break up of a convex-side leading crescent pair in 5% H_2O_2 is shown in SI movie 7 which illustrates the shape-dependent instability of such a small cluster.

- **Changes to the manuscript:** We have now explained our rationale for using reversely swimming particles in the manuscript, see section *Shape Dependence of Crescent Clustering*, paragraph 2, page 7-8.
3. It might be useful to measure the lifetimes of different clusters sizes and discuss the dynamics observed in high n clusters. It is possible that at high n, the clusters are only metastable due to crowding effects and phoretic interactions will likely play a larger role here.
 - We appreciate the referee's idea to study the lifetimes and dynamics of high n clusters. This would require higher particle densities as the fast formation of pairs that rarely merge into clusters dominate at low densities and in our present study we have never observed cluster sizes larger than n=7. At (locally or globally) higher particle densities, we indeed expect that crowding effects and phoretic interactions together with weaker interlocking of crescents attached to the outside of a stable core cluster might lead to the appearance of less-stable configurations and different dynamics. Bubble formation at higher particle densities make it challenging to study higher densities in this system. We are working on implementing a different propulsion method to also access high density phases to study the phase behavior and dynamics, but believe that it exceeds the scope of our current manuscript and have to leave it for future work at the moment.
 4. The authors report experiments over remarkably long periods (90 mins) for this system. Do the authors notice any changes to the observed particle velocities due to fuel depletion effects – especially at higher particle concentrations.
 - We thank the referee for their thoughtful question. While we have never noticed a significant decrease in velocity, we have now measured the velocities of single crescents at the start of the experiment and after 5h in a sample with a particle density comparable to what we report in the main text. We found that the particle velocities agree. This is also in line with earlier reports from our group (Ketzetzi et al. Nature Comm 2022) and with the good match between simulations and experiments.
 - **Changes to the manuscript:** We now mention in the caption of Figure 2 that the average particle velocity does not change significantly even after long times and we have added our experimental results to the SI (see Figure 4).

Minor point:

1. Figure 3 caption contains panel (f) which seems to be missing in the figure.
 - We thank the reviewer for catching this mistake.
 - **Changes to the manuscript:** The caption of Figure 3 has been revised.
2. I also did not find any SI videos connected to Fig4, showing clustering dynamics in $\alpha = \pi/2$ swimmers.
 - We now show videos of small particle clusters for straight rods and crescents with an opening angle of 90° and 260°. Additionally an overview of the different pairs of bend rods with increasing opening angle is now shown in Fig.4.
 - **Changes to the manuscript:** Video S7 - Video S11 were added, see description in the SI. Bright field images of bend-rod-pairs were added to Fig.4 in the main text.

II. REPORT OF THE SECOND REFEREE

Minor comments:

1. I think, the lock-and-key analogy is not appropriate for this particular system as both particles involved are identical (in contrast to the original paper from the David Pine).
 - We thank the referee for their time to review our manuscript and understand their concern regarding the use of the lock-and-key analogy. Our intention in choosing this analogy was to emphasize how strikingly well the interlocking works, most importantly of course in the case of bent rods with an opening angle of π . To eliminate any ambiguity, we revised the text and exchanged the term lock-and-key by the term interlocking.
 - **Changes to the manuscript: The term lock-and-key was replaced by the term interlocking.**
2. The authors in their abstract and in the conclusion state that this work opens up possibilities for “designing targeted two-component drug delivery” that is not demonstrated in the paper at any level and just sounds a bit bombastic.
 - We genuinely hope that our work might contribute to such technology but decided to remove it to avoid any impression of overselling our work.
3. The introduction in general does a good job overviewing dynamics in active systems formed by shape anisotropic particles. I agree with the authors’ statement that there are not many such systems realized experimentally. There is, nevertheless, the recent active system formed by shape-anisotropic (pear-shaped) Quincke rollers that also demonstrate very unusual clustering in response to the activity changes (Nature Communications 11, Article number: 4401 (2020)). Authors might find it useful for the introduction.
 - We thank the referee for their kind words and assessment of the novelty of our work and appreciate the literature suggestion of the referee. The mentioned paper is indeed a good addition to the overview we aimed to give of the field of anisotropic active particles and we have added a reference to it to the introduction.
 - **Changes to the manuscript: The suggested reference has been included in the introduction, see first paragraph of the introduction, page 1.**

III. REPORT OF THE THIRD REFEREE

Major points:

1. First, the authors borrowed the concept of lock and key and use it in active system. This is appealing, but the system presented is about the interlock of two bent rods. I find difficult to relate it to the lock and key model, where shape complementary helps with interactions. In the current system, what is the lock and what is the key? Because the title of the manuscript is “Designing highly efficient lock-and-key interaction”, some introduction and proper reasoning about the lock-and-key interaction is needed. Regardless, I feel it is just the “interlock” of two convex particles rather than “lock and key”.
 - We thank the referee for taking the time to review our manuscript and understand their concern regarding the use of the concept of lock-and-key. Our intention in choosing this analogy was to emphasize how strikingly well the interlocking works, most importantly of course in the case of bent rods with an opening angle of π . To eliminate any ambiguity, we revised the text and exchanged the term lock-and-key by the term interlocking.
 - **Changes to the manuscript: The term lock-and-key was replaced by the term interlocking.**
2. The second concern is about the conclusion that clustering can occur at low particle density, which the authors tried hard to sell. Given the fact that the particle can interlock due to their convex shape, is it obvious that they can form pairs at low concentration (as long as particle can meet in the right orientation)? Once the pair forms, it seems to be irreversible (due to the propulsion). The author compares their system with mobility induced phase separation (MIPS) and few other systems, is it a fair comparison? Of course it is the authors’ credit to report a system that is different, yet I feel the conclusion is somewhat trivial, although the anisotropic shapes indeed plays its role in forming and stabilizing the cluster.

- We respectfully but strongly disagree with the statement that a phenomenon, that to our eye, is far from easily predictable is termed "somewhat trivial". Even if the two-particle interaction might seem straightforward, the subtlety of the clustering mechanism unraveled in this work and the meticulousness of our analysis, which combines experiments and multiple theoretical approaches is not. In fact, as soon as one thinks about the problem the apparent simplicity disappears: our bent-rod colloids are subject to forces - e.g. Brownian, steric and viscous - which are expected to rip the clusters apart. Yet, they persistently cluster, do so much more efficiently than spheres, and follow the same statistics (see e.g. Fig. 4 in the manuscript). Active particles with this shape have not been experimentally explored before, and this is the first experimental demonstration that shape can enhance clustering in systems of active particles. In addition, the precise fashion how the shape and hence interlocking translates into collective behavior is not trivial, and we describe it for a whole class of shapes. We consider neither the statistics (Michaels-Menten etc.) and efficiency of clustering, nor the density-dependency as obvious. Moreover, we find the simple dependence of the clustering efficiency on shape surprising and the identification and quantification of an optimum exciting as it covers a whole class of anisotropic particle shapes. As such our work is the first of its kind and hence clearly novel.

The comparison with MIPS is indeed a fair one, since both phenomena are ultimately related to the break down of detailed balance resulting from directed motion and steric interactions. In bulk systems, this mechanism drives phase-separation at small length scales and thus MIPS. Here we show that the same mechanism can lead to clustering even in the absence of a true bulk phase. To emphasize both the similarity and distinguish it from MIPS, we now refer to this phenomena as motility induced clustering in the text.

There is of course an open question as to how these two phenomena eventually crossover once the density of particles is increased and how shape influences this. This, however, is material for a separate study.

- We have now added the terms "high particle densities" as well as "low particle densities" to the introduction (see paragraph 2, page 1 and paragraph 5, page 2) and are now describing the phenomenon as motility induced clustering (see last paragraph of the introduction). We also added a sentence to the conclusions to clarify the similarity and difference of motility induced clustering and MIPS.

Other comments:

1. When tuning the shape, straight rod is important as a control experiment. Can they also form pairs. The authors should add experiments and analysis about it. In this case, no interlock is available and can be compared properly with one with interlock.

- We had initially decided to not perform such experiments ourselves, because the self-organisation of side-propelling colloidal rods had already been studied by Vutukuri et al. (Soft Matter, 2016, 12, 9657), although not in a similarly quantitative manner as we present here. Therefore, we agree with the referee that active rods can be used as a control group to test the importance of interlocking and have done so now in a new set of experiments. Rods indeed form pairs as well, but are less stable because they can breakup by sliding. Interestingly, the clustering dynamics of the rods also follows a Michaelis Menten kinetics, but, as expected, occurs slower because of the more frequent breakups of the clusters. Both the experimentally measured dynamics and the speed of the clustering agrees with our previous predictions by simulations and the analytical model.

- Changes to the manuscript: The manuscript was modified to include the experimental results for rods ($\alpha = 0$) as well as for crescents with an opening angle of 260° , see section *Shape Dependence of Crescent Clustering*, paragraph 1, page 5-7 as well as Fig 4 and 5.

2. Why do the bent rods move in opposite directions with 1% H₂O₂ and 5% H₂O₂?

- Understanding the mechanism responsible for this reversal in swimming direction is the focus of our ongoing research and we hope to be able to shed light on this matter in the near future. We nevertheless chose to use this system because it allowed us to generate convex-side leading crescents with a perfectly rounded shape. 3D printing the bent rods in the opposite direction would have generated particles with a truncated shape and thus a flat surface, since the particles have to be in contact with the substrate during the printing process. This significantly alters the clustering dynamics.

- We have now included the rationale for using crescents that move in the opposite direction in the manuscript, see section *Shape Dependence of Crescent Clustering*, paragraph 2, page 7-8.
3. Have the author studied particle with a very high density, to study how the interlock influence the cluster formation and dynamics. As the cluster grow larger, it seems that the kind of interlocking for pairs has changed to other configurations and allows more freedom to move around. The so-called “lock and key” interaction has been weakened.
- We appreciate the referee’s idea to study the behavior at higher particle densities. Indeed, we expect that at high particle densities crowding effects together with weaker interlocking of crescents attached to the outside of a stable core cluster might lead to the appearance of less-stable configurations and different dynamics. Our present study, however, focuses on the case of low particle densities where the cluster size never exceeded $n=7$. Bubble formation at higher particle densities make it challenging to study higher densities in this system. We are working on implementing a different propulsion method to also access high density phases to study the phase behavior and dynamics, but believe that it exceeds the scope of our current manuscript and have to leave it for future work at the moment.
- We have added a sentence to the outlook about interesting future directions.
4. While such shape-induced clustering indeed inspire readers, several key points are less discussed: (i) mechanism of the active lock-and-key interaction. The author may provide force analysis on the dimer and trimer in the Video S5. As not all two-particle collision results in formation of the dimer. What kind of kinetic pathway is required. (ii) shape/activity-dependent structures. In current dimers (Figure 2b), the terminal of one particle locates at the center of another particle. If speed or particle shape (opening angle) change, whether the structure changes or even become unstable? These factors may affect the assembly efficiency.
- We appreciate the suggestion of the referee. Regarding (i): it is unclear to us how such a force analysis could be undertaken experimentally. We have attempted to use calibrated optical tweezers to do this but have found that plasmonic resonances at the metallic cap interfere. In addition, the analysis of force measurements on active particles by optical tweezers has to the best of our knowledge not been established yet.
To obtain an answer to the question, we inferred the force from the measured velocity when freely moving, through a force balance of the active force and drag: $F_{active} = F_{drag}$. With $F_{drag} = \zeta * v * L_{arc}$, where ζ is the friction factor, v the velocity and L_{arc} is the arclength. Using the friction factor for bent rods, as mentioned in section 16 of the SI ($\zeta_0 = 1.42 * \pi\eta$, $\zeta_{90} = 1.36 * \pi\eta$, $\zeta_{180} = 1.55 * \pi\eta$, $\zeta_{260} = 1.86 * \pi\eta$), and the measured velocities we can directly calculate the active force as ranging from $F_{active} = 0.04 - 0.1pN$. It is of course correct that not all collisions will lead to interlocking. We can distinguish three cases when two particles interact: (1) direct interlocking, (2) delayed interlocking, i.e. when particles meet with their terminal "legs" upon collision, they form a metastable ring-like structure. Any sliding induced by noise will always result in interlocking in our active system, just with a delay. (3) No interlocking. We already showed the pathway towards interlocking in Figure 2d and in SI Video4 and took these options implicitly into account in our model through the opening angle dependent probability for interlocking. We have furthermore added the above description to the manuscript.
Regarding (ii): With our additional experiments we now have experimental data for four different opening angles and had already previously explored many different opening angles in simulations. The options after collision remain the same: 1) direct interlocking, 2) delayed interlocking, and 3) no interlocking. The shape affects the stability of the formed structure and the probability of interlocking as identified and taken into account by our model and hence the assembly efficiency.
We have not varied the speed experimentally but agree that with an increasing contribution from noise (i.e. lower speed) the stability of the pairs will lower and thus we also expect to lower the assembly efficiency. For the range of speeds observed for different shapes we consistently found the same pair-wise interlocking.
- Changes to the manuscript: The manuscript was modified to include the possible options of collision as well as the range of the active force, see section *Collective Behavior of Active Bent Rods*, paragraph 2-3, page3. A full calculation of the active force and expected rotational coefficient of a crescent-pair can be found in the SI.

5. Brownian motion is negligible for the 10- μm particles, which may benefit the stabilization of clusters. In the presented simulation, noises are ignored with vague description “the noise did not change the behavior significantly.” While extensive biological and synthetic swimmers could be smaller and bear the influence of the Brownian motion. Can the author analyze the influence of the particle size on the “active lock-and-key interaction” by the experiment or the simulation? Admittedly, synthesis of the smaller concave particles could be difficult.

- We agree with the suggestion of the referee and apologize for the vague statement earlier. We had done preliminary simulations with noise to see if the statistics would change but found that this was not significant. Experimentally there is a limit to fabricating much smaller sizes. We can, however, explore this question in simulations. In Sec. 9 of the SI we now show the N_c/N curve for 180° particles as noise increases. The change of number of clustered particles is relatively small when the rotational noise is small. However, for larger values the number decreases significantly.
- Changes to the manuscript: We included the results of the simulations in the SI and mention it in the main text.

REVIEWERS' COMMENTS

Reviewer #1 (Remarks to the Author):

I want to thank the authors for their careful consideration of my comments/suggestions.

The new experimental data provided for particles of different shapes ($\alpha = 0, 90, 270$) and the consistency of this data with simulations makes this manuscript more complete. However I am surprised by the results obtained for the straight rods. Unlike in the case of bent rods, there are no 'nucleation points' for the clusters to form and the rods can easily slide past each other at low densities and I would have expected the data to be similar to convex-leading swimmers. Are there additional hydrodynamic/phoretic mechanisms at play for this geometry (seems to be the case from the videos) leading to metastable states as observed before for rod shaped swimmers (Wykes, Megan S. Davies, et al. *Soft matter* 12.20 (2016): 4584-4589.)? The couplets seem to be particularly stable based on the data in SI Fig 9. This would be different from the conclusions drawn for other shapes from the simulations.

I appreciate that the authors have chosen to move away from the 'lock-and-key' terminology towards 'interlocking' which is more accurate for the mechanism described in this manuscript.

As a note on printing upside down particles, it is understandable that the authors observe a flat surface due to the required contact with the glass substrate. For future consideration, one approach that could work is to use extremely small rod-shaped structures on the glass substrate on which the 'upside down' concave structures are supported. When sonication is used to release the structures from the glass substrate the small rods tend to break away from the main structure and can be filtered out.

While I still maintain reservations about the novelty of the system studied here in light of previous studies – that admittedly have only separately addressed individual aspects of the mechanism described here (but make some of these observations expected), I agree with the authors that the additional experimental data provided and their detailed analysis, and the multiple theoretical/simulation approaches described makes this manuscript quite comprehensive and a useful addition to the active matter literature.

Reviewer #2 (Remarks to the Author):

The authors adequately addressed the comments from the reviewers. I recommend the manuscript for acceptance.

Reviewer #3 (Remarks to the Author):

I have reviewed the revised manuscript. I appreciate it that the authors have tried to address my previous concerns/suggestions. They have replaced the confusing term "lock and key" with "interlocking", and compared their system with those featuring MIPS. They have also added experiments of the rod particles, as control experiments to those by the bent particles.

While I am still not fully convinced about the significance of the findings, the study itself is sound based on a new system; it can inspire more researchers to investigate how anisotropic active colloids may behave differently from spheres and rods.

In conclusion, I can recommend its publication.

One suggestion for consideration: for the title, would it be better if the "designing" is removed. For lock and key colloids, you can design according to complementary shape and then predict the lock and key interaction. Now, in the current state, it is more that you design bent particles with various bending angles, and discover that they can interlock and do so more efficiently when the angle/shape is optimized.

Point by point reply

(Dated: June 13, 2024)

I. REPORT OF THE FIRST REFEREE

I want to thank the authors for their careful consideration of my comments/suggestions. The new experimental data provided for particles of different shapes ($\alpha = 0, 90, 270$) and the consistency of this data with simulations makes this manuscript more complete. However I am surprised by the results obtained for the straight rods. Unlike in the case of bent rods, there are no ‘nucleation points’ for the clusters to form and the rods can easily slide past each other at low densities and I would have expected the data to be similar to convex-leading swimmers. Are there additional hydrodynamic/phoretic mechanisms at play for this geometry (seems to be the case from the videos) leading to metastable states as observed before for rod shaped swimmers (Wykes, Megan S. Davies, et al. *Soft matter* 12.20 (2016): 4584-4589.)? The couplets seem to be particularly stable based on the data in SI Fig 9. This would be different from the conclusions drawn for other shapes from the simulations.

- We thank the referee for appreciating the additional experiments we have made to improve our manuscript. The here reported self-propelled rods can indeed, like convex-side leading crescents, slide past each other after a successful collision. However, due to the curved surface of the bend rods, this effect is faster in the case of the convex-side leading crescents. We note that in all experiments with synthetic microswimmers flow-induced attractive interactions are present which favour clustering and might be what the referee has observed in the supplementary video. The variation in the cluster formation between different shapes, however, is only driven by interlocking, because the rescaled experimental data are in very good agreement with our results from simulations. We therefore concluded that the efficiency of clustering is purely a shape dependent effect.

I appreciate that the authors have chosen to move away from the ‘lock-and-key’ terminology towards ‘interlocking’ which is more accurate for the mechanism described in this manuscript.

As a note on printing upside down particles, it is understandable that the authors observe a flat surface due to the required contact with the glass substrate. For future consideration, one approach that could work is to use extremely small rod-shaped structures on the glass substrate on which the ‘upside down’ concave structures are supported. When sonication is used to release the structures from the glass substrate the small rods tend to break away from the main structure and can be filtered out.

- We thank the referee for their suggestion and will consider it in future particle design steps.

While I still maintain reservations about the novelty of the system studied here in light of previous studies – that admittedly have only separately addressed individual aspects of the mechanism described here (but make some of these observations expected), I agree with the authors that the additional experimental data provided and their detailed analysis, and the multiple theoretical/simulation approaches described makes this manuscript quite comprehensive and a useful addition to the active matter literature.

II. REPORT OF THE SECOND REFEREE

The authors adequately addressed the comments from the reviewers. I recommend the manuscript for acceptance.

- We thank the referee for their time and for recommending the publication of our manuscript.

III. REPORT OF THE THIRD REFEREE

I have reviewed the revised manuscript. I appreciate it that the authors have tried to address my previous concerns/-suggestions. They have replaced the confusing term “lock and key” with “interlocking”, and compared their system with those featuring MIPS. They have also added experiments of the rod particles, as control experiments to those by the bent particles.

While I am still not fully convinced about the significance of the findings, the study itself is sound based on a new system; it can inspire more researchers to investigate how anisotropic active colloids may behave differently from spheres and rods.

In conclusion, I can recommend its publication.

One suggestion for consideration: for the title, would it be better if the "designing" is removed. For lock and key colloids, you can design according to complementary shape and therefore predict the lock and key interaction. Now, in the current state, it is more that you design bent particles with various bending angles, and discover that they can interlock and do so more efficiently when the angle/shape is optimized.

- We thank the referee for recommending the publication of our manuscript. Concerning the title of our manuscript, we think that it fits the essence of what we want to demonstrate, namely that clustering in active particles can be tuned through the shape of the active agents. The design of efficiently interlocking particles was very much intentional when we first developed the bend rod shape.